# The Transcription Factor Nfix Requires RhoA-ROCK1 Dependent Phagocytosis to Mediate Macrophage Skewing during Skeletal Muscle Regeneration

**DOI:** 10.3390/cells9030708

**Published:** 2020-03-13

**Authors:** Marielle Saclier, Michela Lapi, Chiara Bonfanti, Giuliana Rossi, Stefania Antonini, Graziella Messina

**Affiliations:** Department of Biosciences, University of Milan, via Celoria 26, 20133 Milan, Italy; marielle.saclier@unimi.it (M.S.); michela.lapi@unimi.it (M.L.); chiara.bonfanti@unimi.it (C.B.); giuliana.rossi@epfl.ch (G.R.); stefania.antonini@unimi.it (S.A.)

**Keywords:** macrophages, Nfix, skeletal muscle, phagocytosis, RhoA-ROCK1

## Abstract

Macrophages (MPs) are immune cells which are crucial for tissue repair. In skeletal muscle regeneration, pro-inflammatory cells first infiltrate to promote myogenic cell proliferation, then they switch into an anti-inflammatory phenotype to sustain myogenic cells differentiation and myofiber formation. This phenotypical switch is induced by dead cell phagocytosis. We previously demonstrated that the transcription factor Nfix, a member of the nuclear factor I (Nfi) family, plays a pivotal role during muscle development, regeneration and in the progression of muscular dystrophies. Here, we show that Nfix is mainly expressed by anti-inflammatory macrophages. Upon acute injury, mice deleted for Nfix in myeloid line displayed a significant defect in the process of muscle regeneration. Indeed, Nfix is involved in the macrophage phenotypical switch and macrophages lacking Nfix failed to adopt an anti-inflammatory phenotype and interact with myogenic cells. Moreover, we demonstrated that phagocytosis induced by the inhibition of the RhoA-ROCK1 pathway leads to Nfix expression and, consequently, to acquisition of the anti-inflammatory phenotype. Our study identified Nfix as a link between RhoA-ROCK1-dependent phagocytosis and the MP phenotypical switch, thus establishing a new role for Nfix in macrophage biology for the resolution of inflammation and tissue repair.

## 1. Introduction

During their lifetime, tissues encounter physiological and non-physiological damages and an effective regeneration is necessary to make these tissues able to continuously sustain their biological functions. Macrophage-mediated inflammation is a fundamental step for tissue recovery. Macrophages (MPs) are immune cells required for tissue regeneration, as their depletion prevents regeneration of different tissues/organs, such as liver [1], spinal cord [2] and skeletal muscle [3]. In several regenerative processes, two populations of MPs have been described. The first population reaching the damaged tissue is the pro-inflammatory population, also called M1 MPs. Pro-inflammatory MPs secrete pro-inflammatory molecules, being the main actors of dead cell clearance. The second population is composed of the anti-inflammatory MPs, named M2 MPs, that come from pro-inflammatory MPs and are involved in the resolution of the inflammation, wound healing and tissue regeneration or repair [4,5,6,7]. Several studies have shown that an impaired or a precocious phenotypical switch from M1 to M2 MPs results in defective tissue regeneration [1,8,9]. Interestingly, it has been observed that phagocytosis is at the basis of the pro to anti-inflammatory phenotypical switch in MPs [3,8,10,11,12,13]. Although the process of the induction of phagocytosis is well-known (“find-me”, “eat-me” and “don’t eat-me” signals), the molecular and transcriptional pathways between phagocytosis and the phenotypical switch are still unexplored [10,11,14,15].

Interestingly, the interplay between MPs and tissue regeneration has been widely documented in skeletal muscle [7,16,17,18,19,20]. In vertebrates, muscle progenitors originate from pre-somitic and cranial mesoderm. In pre-natal period, two myogenic waves are necessary for muscle establishment: the first forms the basic muscle pattern and is called primary or “embryonic” myogenesis, while the second or “fetal” myogenesis is characterized by muscle maturation and growth [21]. Adult skeletal muscle is able to regenerate thanks to resident stem cells called satellite cells (SCs), located under the basal lamina of myofibers [22]. Upon injury, SCs exit from quiescence, proliferate, differentiate in myoblasts and fuse to reform myofibers [23]. Nuclear factor I X (Nfix) is a transcription factor belonging to the highly conserved DNA-binding nuclear factor one family (Nfi) together with Nfia, Nfib and Nfic [24]. Nfix has a key role in prenatal myogenesis by driving the transcriptional switch from embryonic to fetal myogenesis [25,26]. Nfix is also required for adult myogenesis upon injury, since its absence leads to defect of SC differentiation [27]. Finally, we recently demonstrated that the deletion of Nfix in two mouse models of muscular dystrophy induces a significant morphological and functional amelioration of the pathology by slowing-down muscle regeneration and promoting a switch towards a more oxidative musculature [28].

During muscle regeneration, myogenic cells and MPs closely interact [7]. Soon after injury, activated SCs attract blood monocytes that infiltrate damaged muscle and differentiate in pro-inflammatory MPs that stimulate the proliferation of myoblasts. Then, by removing dead cells, MPs switch to an anti-inflammatory phenotype that sustains myogenic differentiation [3,29]. While MPs are required for muscle regeneration, preventing MPs infiltration in dystrophic disease decreases muscle damage [30]. Thus, depending on a context of acute or chronic injury, MPs adopt a complete opposite function toward muscle cells and environment [31,32].

In this study, we address the role of Nfix in MPs during skeletal muscle regeneration, by using a mouse model in which Nfix is deleted specifically in MPs. We report that mice lacking Nfix in myeloid lineages exhibit a delay of muscle regeneration upon acute injury. We demonstrated that the RhoA-ROCK1-dependent phagocytosis induces Nfix, whose expression is necessary for the acquisition of anti-inflammatory phenotype and thus pro-regenerative properties through myogenic cells. Indeed, during the process of muscle regeneration, in the absence of Nfix, MPs are able to phagocyte, but failed to adopt an anti-inflammatory phenotype necessary for the resolution of inflammation and muscle regeneration.

## 2. Materials and Methods

### 2.1. Animal Models and In Vivo Experimentations

WT, Nfix^fl/fl^ and LysM^CRE^:Nfix^fl/fl^ mice were used in this study. LysM^CRE^:Nfix^fl/fl^ mice were generated, crossing Nfix^fl/fl^ mice obtained from Prof. Richard M. Gronostajski [33] and LysM^CRE^ mice obtained from Dr. Rémi Mounier [8]. All LysM^CRE^:Nfix^fl/fl^ mice analyzed were heterozygous for the LysM^CRE^. Muscle regeneration was realized by the injection of 20 uL of 100 uM cardiotoxin (CTX, Latoxan, L8102) in the *Tibialis anterior* (TA) of 2-month-old mice. For the in vivo analysis of satellite cells and myoblasts proliferation, EdU (5-ethynyl-2’-deoxyuridine) was injected in Nfix^fl/fl^ and LysM^CRE^:Nfix^fl/fl^ mice in intraperitoneal, 12 h before the sacrifice of the mice (100 µL of 6mg/mL EdU solution for 20 g of mouse weight) (Click-iT EdU Imaging Kits Alexa Fluor 594, Thermo Fisher A10044, Paisley, UK). Mice were kept in pathogen-free conditions and all procedures conformed to Italian law (D. Lgs n° 2014/26, implementation of the 2010/63/UE) and approved by the University of Milan Animal Welfare Body and by the Italian Minister of Health.

### 2.2. Isolation of MPs from Skeletal Muscle

Fascia of the TA muscles was removed. Muscles were dissociated and digested in RPMI medium containing 0.2% of collagenase B (Roche Diagnostics GmbH 11088815001) at 37 °C for 1 h and passed through a 70 μm and a 30 μm cell strainer. CD45^+^ cells were isolated using magnetic beads (Miltenyi Biotec 130-052-301) and incubated with FcR blocking reagent (Miltenyi Biotec 130-059-901) for 20 min at 4 °C in PBS 2% FBS. Cells were then stained with Ly6C-PE (eBioscience 12-5932) and CD64-APC (BD Pharmingen 558539) antibodies for 30 min at 4 °C. MPs were analyzed or sorted using a FACS Aria III cell sorter (BD Biosciences) (gating strategy is shown Appendix A). In some experiments, Ly6C^+^ and Ly6C^−^ MPs were cytospined on starfrost (Knitterglaser, Bielefeld, Germany) slides and immunostained.

### 2.3. Histology and Immunofluorescence Analyses

The fascia of TA muscles was removed and the muscles were frozen in liquid nitrogen-cooled isopentane (VWR) and placed at −80 °C until cut. Then, 8 μm-thick cryosections were stained for hematoxylin-eosin (H&E) and immunofluorescence. H&E (Sigma-Aldrich, Saint-Louis, MO 63103, USA) staining was processed according to standard protocols. For immunofluorescence analysis, sections or cells were fixed for 15 min with 4% paraformaldehyde (except for F4/80 and eMyHC staining). Then, samples were permeabilized with 0.5% Triton X-100 (Sigma-Aldrich) in PBS for 10 min and blocked with 4% BSA (Sigma-Aldrich) in PBS at RT for 1 h. Primary antibodies were incubated O/N at 4 °C in PBS. After three washes of 5 min with PBS, samples were incubated with secondary antibodies (1:500, Jackson Laboratory. Fluorochromes used: 488, 594, 546 and 647) and Hoechst (1:500, Sigma-Aldrich) in PBS for 45 min at RT, then washed four times for 5 min with PBS and mounted with Fluorescence Mounting Medium (Dako). For Nfix-F4/80 double immunolabeling, cryosections were labelled with antibodies against F4/80 (1:400, Novus Biologicals NB300-605) overnight at 4 °C and Nfix labelling using (1:200, Novus Biologicals NBP2-15039) the antibody was performed for 2 h at 37 °C. For EdU-Pax7-laminin immunolabelling, after fixation and permeabilization of muscle sections, we followed the manufacturer’s instructions of the Click-iT EdU Imaging Kits Alexa Fluor 594 (Thermo Fisher A10044) to reveal the DNA integrated EdU. Then, for Pax7 immunostaining, antigen retrieval was performed by incubating muscle sections in boiling 10 mM citrate buffer pH6 for 20 min. Muscle sections were then incubated O/N with Pax7 (1:2, Hybridoma, DSHB, Iowa City, IA 52242, USA) and laminin (1:200, Sigma L9393). The other antibodies used were eMyHC (1:2, Hybridome), MyoD (1:50, Santacruz Biotechnology sc-377460), TNFα (1:50, Abcam ab34839), CCL3 (1:500, Abcam ab32609, Cambridge, UK), iNOS (1:25, Novus Biologicals NB300-605, Centennial, CO 80112, USA), CD163 (1:50, Santacruz Biotechnology sc-33560), CD206 (1:50, Bio-Rad MCA2235GA), TGFβ (1:100, Abcam ab64715), Arginase I (1:100, Santacruz Biotechnology, Cambridge, UK).

### 2.4. Bone Marrow Derived MPs (BMDM) Culture

Total mouse bone marrow was obtained by flushing femur and tibiae with DMEM. Cells were cultured in DMEM containing 20% Fetal Bovine Serum (FBS) and 30% of L929 cell line-derived conditioned medium (enriched in CSF-1) for 6 to 7 days. MPs were polarized using 50 ng/mL IFNγ (for M1 polarization) (Peprotech #315-05), 10 ng/mL IL10 (for M2c polarization) (Peprotech #210-10), in DMEM (10% FBS) for 3 days. After washing three times, DMEM serum-free medium was added for 24 h, and supernatants were recovered and centrifuged to obtain macrophage-conditioned medium. For some experiments, cells were directly used for various analyses. In some experiments, DMSO or 10 μM of ROCK inhibitor Y27632 (Santacruz sc-3536) was added on MPs.

### 2.5. Myogenic Progenitor Cells (mpc) Culture

Murine WT myoblast progenitor cells (mpcs) were obtained from TA muscle and cultured in DMEM/F12 (Gibco, Paisley, UK), containing 20% FBS and 2.5 ng/mL of human FGF-basic (Peptrotech, 100-18B). For the proliferation assay, mpcs were seeded at 10 000 cell/cm^2^ on Matrigel (1/10) and incubated for 1 day with macrophage-conditioned medium + 2.5% FBS. Then, cells were incubated with the anti-Ki-67 antibody (1/50, BD Biosciences 550609). For differentiation assay, mpcs were seeded at 30 000 cell/cm^2^ on Matrigel (diluted 1/10 in DMEM/F12) and incubated for 3 days with macrophage-conditioned medium containing 2% horse serum. Then, cells were incubated with a pan-myosin antibody (1:2, Hybridoma).

### 2.6. Phagocytosis Assay

Mpcs were labelled using the CellVue Claret Far Red kit (Sigma-Alrich MinClaret) by following the manufacturer’s instructions (Sigma-Aldrich) and treated with staurosporin at 5 μM for 4 h, in order to induce apoptosis. M1 and M2c polarized MPs were incubated with apoptotic mpcs at a 1:3 ratio for 30 min at 4 °C or 6 h or 16 h at 37 °C. After three PBS washings, MPs were detached using trypsin and a cell scraper and cells were labelled with a CD64-APC (BD Pharmingen 558539) and analyzed by flow cytometry using a FACS Aria III cell sorter (BD Biosciences). The double-positive cells (CD64^+^/Far Red^+^ cells) were phagocytic MPs, whereas the CD64^+^/Far Red^−^ cells were nonphagocytic MPs. To exclude MPs that have bound, but not ingested, apoptotic cells, we subtracted the percentage of double-positive cells observed at 4 °C from the value observed at 37 °C. In some experiments, MPs were treated with 1 μg/mL of cytochalasin D (Sigma-Aldrich C8273), 45 min before adding apoptotic mpcs, and with the added mpcs.

### 2.7. Lentiviral Transduction

BMDM from WT mice were transduced with a lentivirus carrying a scrambled sequence or a shNfix [27]. Transduction was performed in suspension (in DMEM 20% FBS), at a MOI of 10 and in the presence of Polybrene (8 μg/mL, Sigma-Aldrich). After O/N incubation, the medium was changed and cells were treated with puromycin (2 μg/mL, Sigma-Aldrich).

### 2.8. RNA Extraction and qRT-PCR

RNA was isolated from the sorted apoptotic mpcs, non-phagocyted and phagocyted MPs, by using TRIzol Reagent (Invitrogen 15596026, Bleiswijk, Netherlands), according to the manufacturer’s instructions. RNA was quantified using a NanoPhotoneter (Implen). For retro-transcription, 500 ng of RNA was used with the iScript Reverse Transcription Supermix for RT-quantitative qPCR (Bio-Rad 1708840). For qRT PCR, cDNA was diluted 1:10, and 5 μL of the diluted cDNA was loaded in a total volume of 20 μL (SYBR Green Supermix (Bio-Rad 172-5124) and run on the Bio-RAD CFX Connect Real-Time System. The relative quantification of gene expression was determined by the comparative CT method, and normalized to Cyclophiline A. Primers used were: Nfix for CTGGCTTACTTTGTCCACACTC; Nfix rev CCAGCTCTGTCACATTCCAGAC; Myogenin for CTGGGGACCCCTGAGCATTG; Myogenin rev ATCGCGCTCCTCCTGGTTGA; Cyclo A for GTGACTTTACACGCCATAATG; Cyclo A rev ACAAGATGCCAGGACCTGTAT.

### 2.9. Protein Extraction and Western Blot

Protein extracts were obtained from cultured MPs lysed using RIPA buffer (10 mM Tris-HCl pH 8.0, 1 mM EDTA, 1% Triton-X, 0.1% sodium deoxycholate, 0.1% sodium dodecylsulphate (SDS), 150 mM NaCl, in deionised water), plus protease and phosphatase inhibitors for 30 min on ice. Then, samples were centrifuged at 11.000× *g* for 10 min at 4 °C, and the supernatants collected for protein quantification (DC Protein Assays Bio-Rad 5000111). 40 μg protein of each sample were denatured at 95 °C for 5 min using SDS PAGE sample-loading buffer (100 mM Tris pH 6.8, 4% SDS, 0.2% bromophenol blue, 20% glycerol, 10 mM dithiothreitol) and loaded into 8% SDS acrylamide gels. After electrophoresis, the protein was blotted into nitrocellulose membranes (Protran nitrocellulose transfer membrane; Whatman) for 2 h at 70 V at 4 °C. Membranes were then blocked for 1 h with 5% milk in Tris-buffered saline, plus 0.02% Tween20 (Sigma-Aldrich). Membranes were incubated with the primary antibodies O/N at 4 °C, using the following antibodies: rabbit anti-Nfix (1:1000, Novus Biologicals NBP2-15039), mouse anti-vinculin (1:2500, Sigma-Aldrich V9131), rabbit anti-MYPT1 phosphorylated in Thr696 (1:500, SantaCruz Biotechnology sc-17556-R), and rabbit anti-Tot MYPT1 (1:500; SantaCruz Biotechnology, H-130). After incubation with the primary antibodies, the membranes were washed 3 times for 5 min and incubated with the secondary antibodies (1:10,000, IgG-HRP, Bio-Rad) for 45 min at RT, and then washed again 5 times for 5 min. Bands were revealed using ECL detection reagent (ThermoFisher), with images acquired using the ChemiDoc MP system (Bio-Rad). The Image Lab software was used to measure and quantify the bands of at least three independent western blot experiments. The obtained absolute quantity was compared with the reference band (Vinculin) and expressed in the graphs as normalized volume (Norm. Vol. Int.).

### 2.10. Image Acquisition and Quantification

Images were acquired with an inverted microscope (Leica-DMI6000B) equipped with Leica DFC365FX and DFC400 cameras and 20× and 40× magnification objectives. Necrotic myofibers were defined as pink pale patchy fibers, and phagocyted myofibers were defined as pink pale fibers invaded by basophilic single cells (MPs). For the quantification of CSA, analyses were done on damaged TA, which presented at least 75% of injured muscle. At least 8 pictures in different fields were taken and at least 500 myofibers were analyzed. For each condition of each experiment, at least 8 fields chosen randomly were counted. The number of labelled MPs or mpcs was calculated using the cell tracker in ImageJ software and expressed as a percentage of total MPs or mpcs. Fusion index was the number of nuclei within myotubes divided by the total number of nuclei.

### 2.11. Statistical Analysis

All data shown in the graph are expressed as mean ± SEM. All experiments were performed using at least three different cultures or animals in independent experiments. A statistical analysis was performed using two-tailed unpaired Student’s t-Test, one-way ANOVA or two-way ANOVA. * *p* < 0.05; ** *p* < 0.01; *** *p* < 0.001; confidence intervals 95%, alpha level 0.05.

## 3. Results

### 3.1. Nfix is Expressed by Anti-Inflammatory MPs

To understand if the transcription factor Nfix could be involved in MP function, we first analyzed Nfix expression in MPs during normal muscle regeneration in WT mice. We induced muscle injury by cardiotoxin (CTX) injection in the *Tibialis Anterior* (TA) and looked at the number of MPs (F4/80^+^ cells) positive for Nfix (Figure 1a). While the number of Nfix-positive MPs was identical between day two (D2) and day four (D4) after injury, we observed an increase of MPs expressing Nfix at D7 after CTX injection (Figure 1a). During muscle regeneration, two populations of MPs are present in the damaged tissue. First, the Ly6C^+^ pro-inflammatory MPs appear and then, they switch into Ly6C^−^ anti-inflammatory population [3,9,29,34]. Thus, we asked if Nfix could be expressed by one subset of MPs. We sorted MPs after CTX injury at different time points and looked at Nfix expression by immunolabelling. We firstly enriched for CD45^+^ cells (mainly composed of MPs and neutrophils) using magnetic beads and then we used the known markers CD64 and Ly6C to separate the pro-inflammatory (CD64^+^/Ly6C^+^ cells) to the anti-inflammatory MPs (CD64^+^/Ly6C^−^ cells) (Appendix A). We observed that the percentage of pro-inflammatory MPs expressing Nfix does not change over the time of the regeneration (Figure 1b). On the contrary, the percentage of CD64^+^/Ly6C^−^ cells positive for Nfix always increased over time (Figure 1b). We also isolated BMDMs (bone marrow derived MPs) from WT mice and polarized them in pro- or anti-inflammatory phenotype. We observed that anti-inflammatory MPs express more Nfix compared to pro-inflammatory MPs (Figure 1c). Therefore, we can conclude that in both in vitro and in vivo analyses, Nfix is more expressed by anti-inflammatory MPs.

### 3.2. Nfix Expression in MPs is Essential for Muscle Regeneration

We previously demonstrated that Nfix is necessary for the correct differentiation of SCs and, as a consequence, muscle regeneration [27]. In order to understand whether Nfix expression by anti-inflammatory MPs is required for this process, we generated the LysM^CRE^:Nfix^fl/fl^ mice to obtain an animal model deleted for Nfix only in the myeloid line. Once the proper deletion of Nfix in BMDM and CD45^+^ infiltrated cells two days after muscle injury was verified (Appendix A), we evaluated the overall phenotype of this new animal model, with particular interest in skeletal muscle morphology at one and two months of life (Appendix A). No differences were observed between the Nfix^fl/fl^ control and the LysM^CRE^:Nfix^fl/fl^ mice model in terms of general mouse growth, TA/mouse weight and myofiber size (CSA: Cross Sectional Area) (Appendix A). Additionally, we did not observe significant differences in the number of resident MPs expressing Nfix between the Nfix^fl/fl^ and the LysM^CRE^: Nfix^fl/fl^ mice (Appendix A). Therefore, the specific deletion of Nfix in MPs in the LysM^CRE^: Nfix^fl/fl^ mice does not influence the general development of the mice. Notably, in the LysM^CRE^: Nfix^fl/fl^ animals resident MPs expressed Nfix similarly to control mice, meaning that the expression of LysM is not required for the establishment of resident MPs.

We then induced muscle injury in control and LysM^CRE^: Nfix^fl/fl^ mice by CTX injection in TA and we quantified the number of necrotic, phagocyted and regenerating myofibers (Figure 2a,b). Two days after injury, all the myofibers of Nfix^fl/fl^ and LysM^CRE^: Nfix^fl/fl^ mice were in necrosis or phagocyted (Figure 2b). At D4 in the control mice, some necrotic and phagocyted myofibers were present, but already 60% of the fibers were centronucleated (Figure 2b). On the contrary, in the LysM^CRE^:Nfix^fl/fl^ mice, we observed a significant decrease in the percentage of centronucleated myofibers (−31%) (Figure 2b). While at D7, almost all myofibers were in regeneration in the control mice, the LysM^CRE^:Nfix^fl/fl^ mice still exhibited an increase of the percentage of necrotic and phagocyted myofibers and a decrease of centronucleated myofibers (+282%, +150% and −30% respectively), suggesting a delay in the process of muscle regeneration in the absence of Nfix (Figure 2b). We also quantified the CSA of myofibers at D14 and D28 after CTX injury and, in both cases, we observed a decrease of the caliber of myofibers in the LysM^CRE^:Nfix^fl/fl^ compared to the Nfix^fl/fl^ mice, due to a decrease of the number of big myofibers and an increase of small myofibers (Appendix A and Figure 2c). These results demonstrated that the expression of Nfix by MPs is necessary for the proper process of muscle regeneration upon acute injury.

### 3.3. Nfix is Required for MP Phenotypical Switch In Vivo and In Vitro

Defects of muscle regeneration due to MP dysfunction are usually linked to a defect of phenotype acquisition [8,9,35,36]. Thus, we looked at the switch from pro- to anti-inflammatory phenotype after CTX injury at different time points by FACS (Figure 3a). First, we did not observe any differences in neutrophils and MPs infiltration between the two mouse models at all time points analyzed (Appendix A). At two days after injury, a majority of Ly6C^+^ pro-inflammatory MPs was observed in the control Nfix^fl/fl^ mice (Figure 3b). Then, at D4 and D7, the ratio between Ly6C^+^/Ly6C^−^ MPs decreased due to the switch from pro- to anti-inflammatory phenotype (Figure 3b). Interestingly, the ratio between Ly6C^+^/Ly6C^−^ in the LysM^CRE^:Nfix^fl/fl^ mice was always higher compared to the Nfix^fl/fl^ control, meaning that Nfix is necessary for the switch from the pro- to anti-inflammatory phenotype (Figure 3b).

We also silenced Nfix in WT BMDM (Bone Marrow Derived MPs) by using a lentiviral vector carrying a small hairpin RNA targeting Nfix (shNfix), or a scrambled sequence as a control (shScramble) [27]. The decrease of Nfix expression in shNfix infected MPs was confirmed by qRT-PCR (Appendix A). We polarized transduced MPs in pro-inflammatory (M1) and anti-inflammatory (M2c) phenotype (with IFN-γ and IL-10, respectively), and we looked at the expression of several pro- and anti-inflammatory markers by immunofluorescence. As expected, M1 shScramble MPs expressed significantly more TNFα, iNOS and CCl3 pro-inflammatory markers than M2c shScramble MPs (Figure 3c). Conversely, M2c shScramble MPs expressed more ArgI, TGFβ, CD163 and CD206 anti-inflammatory markers than M1 shScramble MPs (Figure 3c). Interestingly, in the absence of Nfix we observed an increase of pro-inflammatory markers (except for CCl3) in MPs polarized to M2c phenotype (Figure 3c). We also observed a decrease of MPs positive for anti-inflammatory markers in the polarized M2c MPs lacking Nfix (Figure 3b). These results clearly show that Nfix is necessary for the proper adoption of an anti-inflammatory phenotype and that, without Nfix, MPs remain in a pro-inflammatory status.

### 3.4. Nfix is Required for Macrophage Function on Mpcs In Vivo and In Vitro

Since depending on their phenotype MPs act differentially on WT myogenic progenitor cells (mpcs), we set experiments of conditioned medium (CM) coming from pro- or anti-inflammatory MPs [37]. We added CM coming from BMDM derived from Nfix^fl/fl^ and LysM^CRE^:Nfix^fl/fl^ mice on proliferating or differentiating mpcs. We looked at mpc proliferation by mean of Ki67 staining and at their differentiation by the quantification of the fusion index. As expected, CM coming from M1 Nfix^fl/fl^ MPs stimulated the proliferation of mpcs, while M2c CM had no effect (Figure 4a and Appendix A). Similarly, CM coming from M1 LysM^CRE^:Nfix^fl/fl^ MPs stimulated at the same extent the proliferation of mpcs (Figure 4a and Appendix A). Interestingly, CM from M2c LysM^CRE^:Nfix^fl/fl^ MPs stimulated the proliferation of mpcs as M1 CM does (Figure 4a and Appendix A), whereas CM coming from both Nfix^fl/fl^ and LysM^CRE^:Nfix^fl/fl^ M1 MPs had no effect on the fusion index of mpcs (Figure 4b and Appendix A). While M2c Nfix^fl/fl^ MPs CM increased the fusion of mpcs compared to M1 Nfix^fl/fl^ CM, the CM from M2c LysM^CRE^:Nfix^fl/fl^ MPs lost its pro-fusion effect (Figure 4b and Appendix A ). We also investigated if the lack of Nfix in MPs affects myogenic cells in vivo. To quantify mpcs proliferation, we injured, using CTX, the TA of both Nfix^fl/fl^ and LysM^CRE^:Nfix^fl/fl^ mice and we injected EdU 12 h before sacrifice. The proliferation of SCs (EdU^+^/Pax7^+^ cells) was identical between the two models at all the time points analyzed (Appendix A). On the contrary, the percentage of EdU^+^/MyoD^+^ cells at D2 and D4 was higher in the LysM^CRE^:Nfix^fl/fl^ mice compared to Nfix^fl/fl^ mice (Figure 4c). To quantify the number of newly formed myofibers, we performed an immunofluorescence for eMyHC (embryonic Myosin Heavy Chain) on both injured mouse models (Figure 4d). At D4, the percentage of myofibers positive for eMyHC was around 80%, meaning that almost all myofibers were formed de novo (Figure 4d). At D7, 20% of the Nfix^fl/fl^ myofibers were positive for the eMyHC and only 6.5% at D14 (Figure 4d). On the contrary, at D7 and D14 we observed an increase in the number of eMyHC^+^ myofibers in the LysM^CRE^:Nfix^fl/fl^ mice (34.7% and 15.8% respectively) (Figure 4d). To conclude, Nfix is necessary to MPs to adopt an anti-inflammatory phenotype and, consequently, function. The defect of the phenotypical switch due to the absence of Nfix results in a persistence of pro-inflammatory MPs in the injured muscle, leading to a continuous proliferation of MyoD^+^ cells. The absence of anti-inflammatory MPs induces a delay in the differentiation of new myofibers and, therefore, in the proper muscle regeneration.

### 3.5. Phagocytosis Induces the Expression of Nfix

It has been shown in literature that the phagocytosis of apoptotic cells is the process driving the switch from pro- to anti-inflammatory phenotype, and several studies have demonstrated that MPs presenting a switch defect have a decrease of phagocytic capacity [8,9,35,38]. So, we decided to investigate whether the phagocytosis is altered in LysM^CRE^:Nfix^fl/fl^ MPs compared to Nfix^fl/fl^ MPs. Primary mpcs previously labelled with CellVue-647 were induced to apoptosis and added on Nfix^fl/fl^ or LysM^CRE^:Nfix^fl/fl^ MPs. After 6 h in culture, we used a CD64 antibody to discriminate MPs from mpcs: apoptotic mpcs are CellVue-647^+^/CD64^-^, non-phagocytic MPs are CellVue-647^−^/CD64^+^ and phagocytic MPs are CellVue-647^+^/CD64^+^ (Appendix A). Surprisingly, we did not observe any difference in the phagocytic capacity of Nfix^fl/fl^ and LysM^CRE^:Nfix^fl/fl^ MPs (Figure 5a). Interestingly, while Nfix^fl/fl^ MPs in contact with apoptotic mpcs adopted an anti-inflammatory phenotype (Appendix A), LysM^CRE^:Nfix^fl/fl^ MPs failed to switch from a pro- to anti-inflammatory phenotype (Appendix A). Thus, we hypothesized that phagocytosis could induce Nfix expression. To answer to this question, we did the same experiment of phagocytosis using WT MPs and we sorted MPs according to their phagocytic capability (phagocytic and non-phagocytic WT MPs, respectively) (Figure 5b). To verify that no mpcs were sorted with MPs, we first analyzed the expression of myogenin in apoptotic mpcs and in both non-phagocytic and phagocytic MPs. Apoptotic mpcs highly expressed myogenin compared to non-phagocytic and phagocytic MPs and no differences in myogenin expression was observed between the two populations of MPs (Appendix A). Interestingly, we observed an increase of Nfix expression and MPs positive for Nfix in phagocytic MPs compared to the non-phagocytic ones (Figure 5b). On the contrary, treatment of MPs with cytochalasin D (an inhibitor of phagocytosis) prevents the increase of Nfix positive MPs (Figure 5c and Appendix A).

We recently demonstrated that the inhibition of the RhoA-ROCK1 pathway induces Nfix expression in fetal myoblasts and numerous studies have shown that the inhibition of RhoA-ROCK1 increases the phagocytosis, while its stimulation prevents phagocytosis [39,40,41,42,43]. Thus, we treated WT MPs with Y27632, an inhibitor of ROCK1, and after 1 h of treatment, the phosphorylation of ROCK1-target Mypt decreased, meaning that the inhibition of RhoA-ROCK1 pathway was effective (Appendix A). After 16 h of treatment, Y27632-treated WT MPs exhibited an increase of Nfix protein (Figure 5d); most importantly, this increase led to a reduction of pro-inflammatory markers and an increase of anti-inflammatory markers (Figure 5e). On the contrary, this switch through an anti-inflammation phenotype did not occur in LysM^CRE^:Nfix^fl/fl^ MPs treated with Y27632 (Figure 5f). These results show that RhoA-ROCK-dependent phagocytosis induces the expression of Nfix, which in turn is necessary to promote the phenotypical switch of MPs from pro- to anti-inflammatory.

## 4. Discussion

Skeletal muscle regeneration requires specific temporal steps for the efficacious tissue reconstruction and MPs are the immune cells that are necessary to this process [3,29]. Previous work from our group demonstrated that *Nfix* null mice exhibit a delay of muscle regeneration, due to a defect of SC differentiation [27]. In this study, we show that the transcription factor Nfix is also expressed by MPs and that mice lacking Nfix in the myeloid lineage have defects in muscle regeneration upon acute injury. We observed that Nfix is preferentially expressed by anti-Ly6C^−^ MPs and that its expression increases in time with the progression of the regenerative process. Using an shNfix strategy, we observed that M2c MPs silenced for Nfix express higher levels of pro-inflammatory markers (TNFα and Cox2), while they express lower levels of anti-inflammatory markers (CD163, CD206, ArgI and TGFβ) than polarized M2c control MPs. Importantly, we observed in vitro that LysM^CRE^:Nfixf^l/fl^ M2c MPs act as M1 MPs on myogenic cells: they stimulate myogenic proliferation and are unable to sustain myogenic differentiation. These two features also occur in vivo since without Nfix, MPs exhibit a defect of phenotypical switch from pro-Ly6C^+^ to anti-Ly6C^−^ MPs, and within the injured muscle, there is a persistence of myoblast (MyoD^+^ cells) proliferation and a delay of newly formed myofibers (eMyHC^+^ myofibers). Previous studies showed that the temporal window of the phenotype skewing is a critical step of an effective regeneration. The switch defect [8,35,36] or early appearance of anti-inflammatory MPs impairs muscle regeneration [9]. In line with this evidence, the impairment in the acquisition of an anti-inflammatory phenotype in MPs lacking Nfix leads to a muscle regenerative delay.

So far, the function of Nfix was mainly analyzed in myogenic and neural cells during both development and adult life [25,27,44,45,46]. Recently, Nfix was also shown to play a positive role in the survival of hematopoietic stem and progenitor cells (HSPC) [47], but also to be involved in the fate decision between early B lymphopoiesis and myelopoiesis from blood HSPC [48]. During development, yolk sac gives rise to tissue resident MPs and fetal liver to HSPC, from which blood monocytes and damaged-infiltrating MPs are derived [49]. In our experiments, no differences in the number of infiltrating MPs between control and LysM^CRE^:Nfix^fl/fl^ mice were observed, meaning that the delay observed is due to a defect of macrophage features within the damaged muscle, but not in terms of failed HSPC development.

Little is known about Nfix up-stream regulation, but recently our laboratory identified ERK and RhoA-ROCK1 pathways as, respectively, positive and negative regulators of Nfix expression in pre-natal muscle development. The inhibition of RhoA-ROCK1 induces Nfix expression, promoting myoblasts fusion which is a reflect of myogenesis progression [39]. In MPs, the inhibition of the RhoA-ROCK1 pathway increases the clearance of dead cell phagocytosis, while the constitutive activation of RhoA reduces their phagocytic capacity [40,41]. Importantly, phagocytosis is the process responsible for the induction of the pro- to anti-phenotypical switch in MPs. While numerous studies investigated the mechanisms involved in the progression or inhibition of phagocytosis, how apoptotic cells attract MPs and how MPs recognize them is still unknown [10,14,15,50,51]. In our study, phagocytosis of LysM^CRE^:Nfix^fl/fl^ MPs was not impaired compared to control cells. We observed that upon phagocytosis, MPs exhibit an increase in Nfix expression and, conversely, the inhibition of phagocytosis, by using the inhibitor of actin polymerization cytochalasin D, prevents Nfix expression. The stimulation of phagocytosis using the ROCK1 inhibitor Y27632 increases Nfix protein, therefore decreasing pro-inflammatory markers and increasing anti-inflammatory markers in WT MPs. On the contrary, either after phagocytosis or after ROCK1 inhibitor treatment, we observed that MPs lacking Nfix do not have a decrease of pro-inflammatory markers and an increase of anti-inflammatory markers. Thus, the inhibition of the RhoA-ROCK1 pathway induces phagocytosis, leading to Nfix expression that, in turn, drives the MP phenotypical switch.

This study is particularly relevant in light of the recent role for Nfix in muscular dystrophies (MDs)[28]. We indeed demonstrated that the lack of Nfix in two different dystrophic animal models improves both morphological and functional parameters associated to the disease, by promoting a more oxidative musculature and by slowing down muscle regeneration [28]. Different studies have shown that the improvement of dystrophies correlates with a decrease of MPs infiltration [30,32]. While MPs are necessary for muscle regeneration upon acute injury, they are deleterious in the case of chronic injury. Indeed, in muscle myopathies, as in several chronic injured pathologies, MPs are at the origin of fibrosis [4,32,52,53]. In the context of acute injury regeneration, pro-inflammatory MPs secrete TNFα that stimulates myoblast proliferation and fibroblast apoptosis, whereas anti-inflammatory MPs secrete TGFβ that promotes myoblast fusion, but also fibroblast proliferation [29,54]. In muscular dystrophies, numerous studies demonstrated that the fibrosis establishment is linked to an over-activation of the TGFβ pathway that stimulates collagen expression by fibroblasts, and in a dystrophic context, more than 75% of MPs express TGFβ [54,55,56,57,58,59,60,61,62]. Thus, in muscle tissue, MPs closely interact with fibroblasts, promoting normal matrix reformation upon acute injury and fibrosis in chronic injury. With this study, we identified Nfix as a new actor of MPs, demonstrating that Nfix is the link between phagocytosis and the phenotypical switch, a necessary step for the resolution of inflammation and tissue repair. Increasing knowledge about signals and factors controlling MP phenotype and, consequently, functions, will help us to understand and control their function in fibrotic pathologies.

## Figures and Tables

**Figure 1 cells-09-00708-f001:**
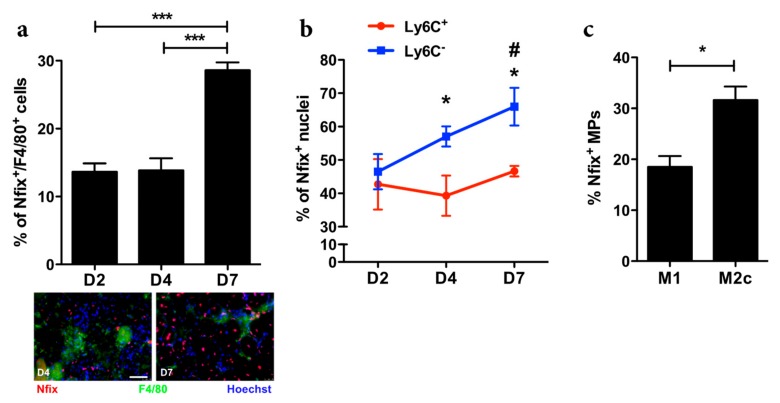
Nfix is mainly expressed by anti-inflammatory MPs. (**a**) Percentage of F4/80^+^ MPs positive for Nfix in *Tibialis Anterior* muscles (TA) of WT mice injected by CTX at D2, D4 and D7, post-injury. Immunostaining for F4/80 (green), Nfix (red) and DAPI (blue) at D4 and D7 after CTX injection; (**b**) Percentage of Ly6C^+^ and Ly6C^-^ sorted MPs positive for Nfix in TA muscles of WT mice injected by CTX at D2, D4 and D7 post-injury; (**c**) Percentage of Nfix^+^ MPs after M1 and M2c polarization (with IFNγ and IL10, respectively). * *p* < 0.05; *** *p* < 0.001; for (b) * *p* < 0.05 Ly6C+ vs. Ly6C^+^ at D4 and D7; # *p* < 0.05 Ly6C^−^ D7 vs. D2. Results are means ± SEM of at least three independent experiments. Scale bar = 50 μm.

**Figure 2 cells-09-00708-f002:**
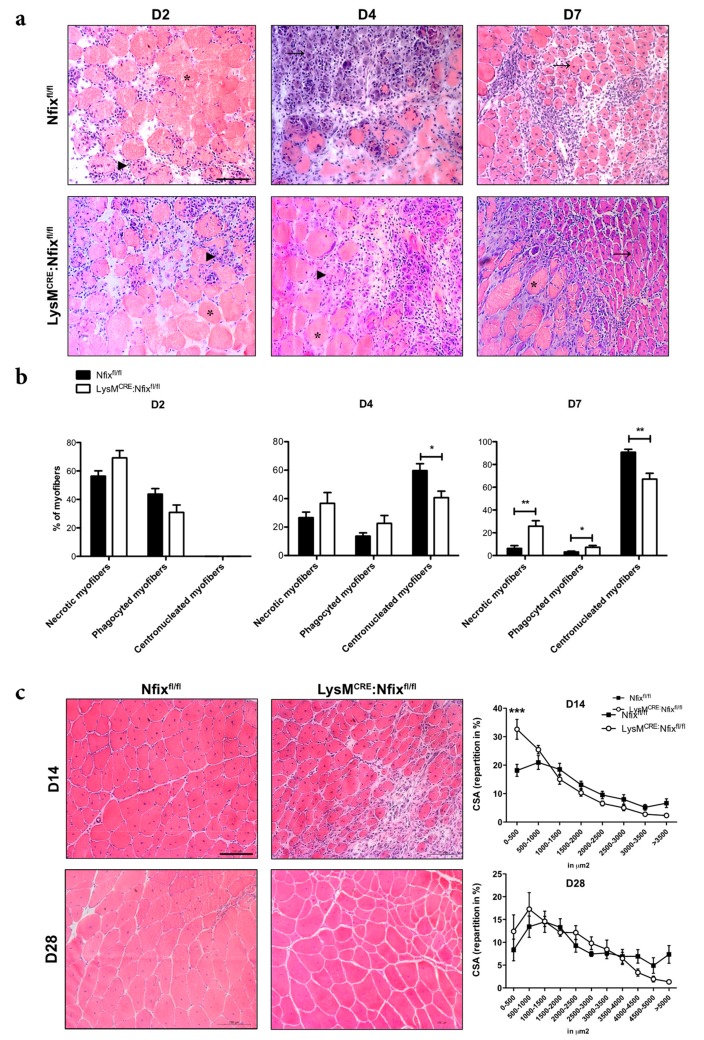
Lack of Nfix in MPs induces a delay of skeletal muscle regeneration. (**a**) Hematoxylin-eosin staining of Nfix^fl/fl^ and LysM^CRE^:Nfix^fl/fl^ TA muscles injected by CTX at D2, D4, D7 postinjury; (**b**) Quantification of necrotic (asterisk), phagocyted (arrowhead) and centrally-nucleated (arrow) myofibers, expressed as percentage out of total myofibers; (**c**) Hematoxylin-eosin staining of Nfix^fl/fl^ and LysM^CRE^:Nfix^fl/fl^ TA muscles injected by CTX at D14 and D28 postinjury and repartition in percentage of the cross-sectional area (CSA). * *p* < 0.05, ** *p* < 0.01, *** *p* < 0.001. Results are means ± SEM of at least three independent experiments. Scale bar = 100 μm.

**Figure 3 cells-09-00708-f003:**
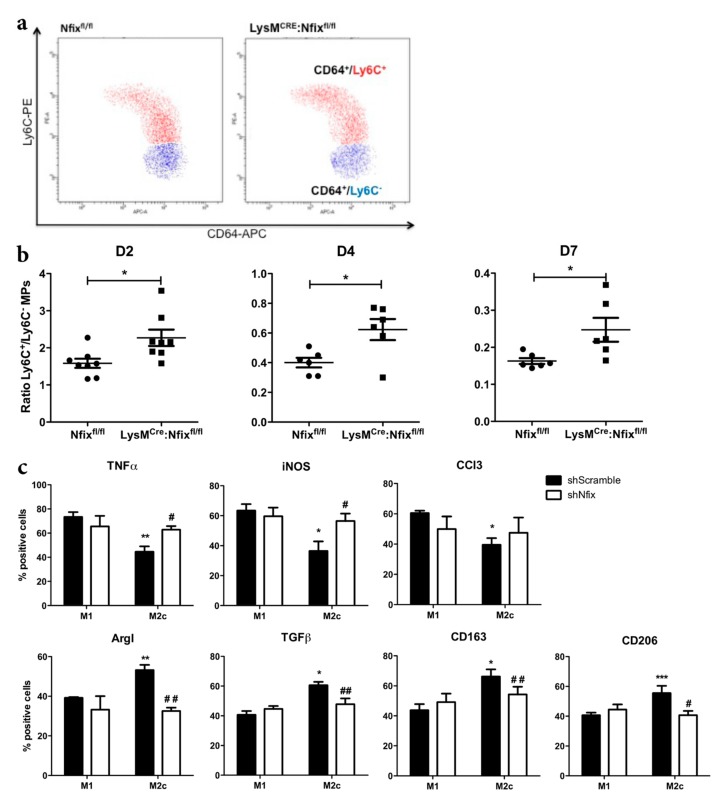
MPs lacking Nfix are unable to adopt an anti-inflammatory phenotype in vivo and in vitro. (**a**) Representative FACS (Fluorescence-Activated Cell Sorting) gate of pro- and anti-inflammatory CD64^+^ MP populations in TA of Nfix^fl/fl^ and LysM^CRE^:Nfix^fl/fl^ mice at D2 after CTX injection. (CD64^+^/Ly6C^+^ and CD64^+^/Ly6C^−^ respectively); (**b**) Ratio of Ly6C^+/^Ly6C^−^ MPs sorted from TA of Nfix^fl/fl^ and LysM^CRE^:Nfix^fl/fl^ mice at D2, D4 and D7, after CTX injection; (**c**) WT BMDM (Bone Marrow Derived Macrophages) were transduced by shScramble and shNfix lentiviral vectors and then polarized into M1 and M2c MPs with IFNγ and IL10 treatment, respectively. MPs were immunolabeled for pro-inflammatory markers (TNFα, iNOS and CCl3) and anti-inflammatory markers (ArgI, TGFβ, CD163 and CD206). The number of positive cells is expressed as percentage out of total cells. * *p* < 0.05; ** *p* < 0.01; *** *p* < 0.001 vs. shScramble M1 MPs. ^#^
*p* < 0.05, ^##^
*p* < 0.01, ^###^
*p* < 0.001 vs. shScramble M2c MPs. Results are means ± SEM of at least three independent experiments.

**Figure 4 cells-09-00708-f004:**
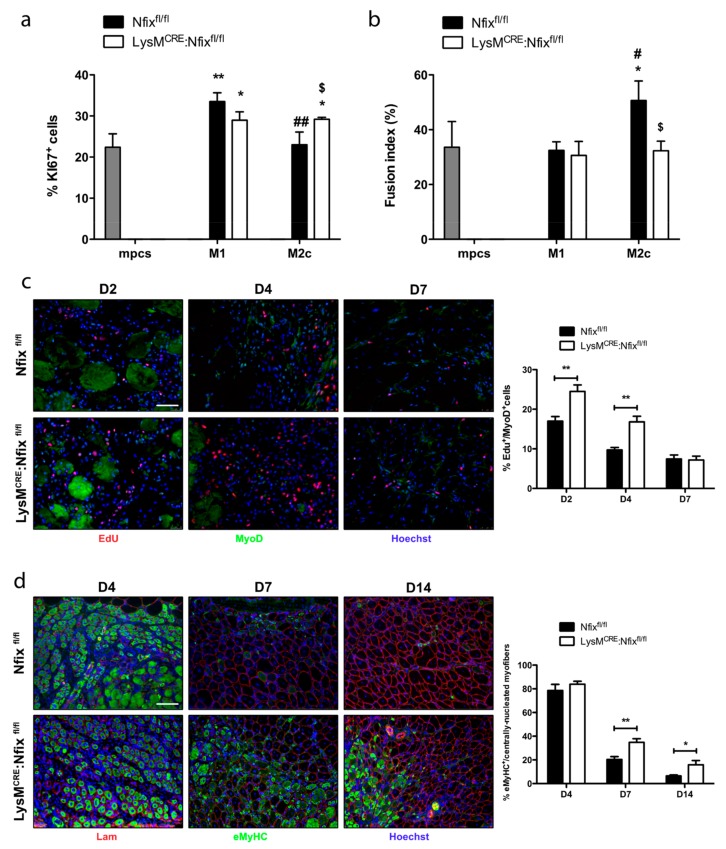
M2 MPs lacking Nfix display M1 MP features on myoblasts in vitro and in vivo. (**a**) Conditioned medium of M1 or M2c polarized Nfix^fl/fl^ and LysM^CRE^:Nfix^fl/fl^ BMDM was added on mpcs, and after 24 h, mpc proliferation was measured as a percentage of Ki67^+^ cells; (**b**) Conditioned medium of M1 or M2c polarized Nfix^fl/fl^ and LysM^CRE^:Nfix^fl/fl^ BMDM was added on mpcs and after 72 h, mpcs fusion index was calculated after sarcomeric MyHC staining (% of MyHC^+^nuclei into myotubes out of the total nuclei). * *p* < 0.05, ** *p* < 0.01 vs. mpcs. ^#^
*p* < 0.05, ^##^
*p* < 0.01 versus M1 Nfix^fl/fl^. ^$^ p < 0.05 versus same Nfix^fl/fl^ polarization; (**c**) Immunostaining for EdU (red), MyoD (green) and Hoechst (blue) of Nfix^fl/fl^ and LysM^CRE^:Nfix^fl/fl^ TA injected by CTX, at D2, D4 and D7 post-injury and quantification of EdU^+^/MyoD^+^ cells. EdU was injected in Nfix^fl/fl^ and LysM^CRE^:Nfix^fl/fl^ mice 8 h before sacrifice; (**d**) Immunostaining for Lam (red), eMyHC (green) and Hoechst (blue) of Nfix^fl/fl^ and LysM^CRE^:Nfix^fl/fl^ TA injected by CTX, at D4, D7 and D14 post-injury and quantification of eMyHC^+^/centrally-nucleated myofibers. * *p* < 0.05, ** *p* < 0.01 Results are means ± SEM of at least three independent experiments. Scale bar = 50 μm.

**Figure 5 cells-09-00708-f005:**
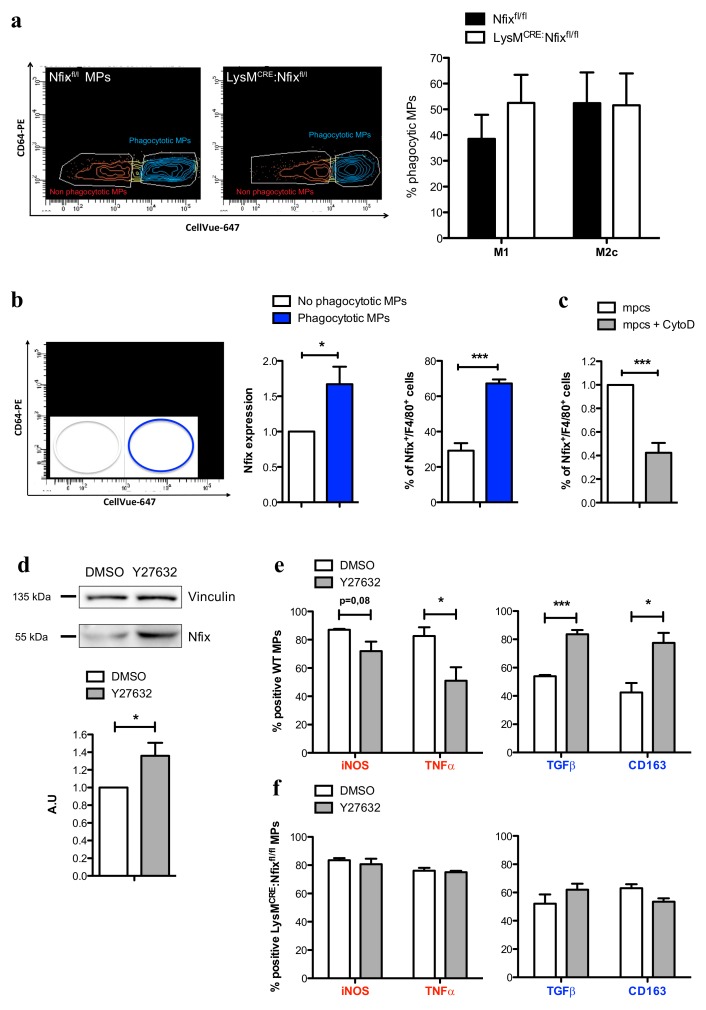
Nfix is expressed after phagocytosis and drive MP phenotypical switch. (**a**) Phagocytosis assay of M1 and M2c Nfix^fl/fl^ and LysM^CRE^:Nfix^fl/fl^ MPs cocultured 8h with apoptotic mpc. Representative FACS gate of phagocytotic M2c Nfix^fl/fl^ and LysM^CRE^:Nfix^fl/fl^ MPs (CD64^+^CellVue^+^) and percentage of phagocytotic M1 and M2c MPs coming from Nfix^fl/fl^ and LysM^CRE^:Nfix^fl/fl^ BMDM; (**b**) WT MPs were cocultured 16h with apoptotic mpcs. Representative FACS gate of non-phagocytotic (CD64^+^CellVue^−^) and phagocytotic (CD64^+^CellVue^+^) WT MPs. Quantification of Nfix expression realized by RT-qPCR on sorted non-phagocytotic and phagocytotic WT MPs and quantification of MPs positive for Nfix (Nfix^+^/F4/80^+^) realized by IF on non-phagocytotic and phagocytotic WT MPs; (**c**) WT MPs were cocultured for 16 h with apoptotic mpcs, with or without addition of Cytochalasin D. Quantification of F4/80^+^ MPs were positive for Nfix on a total of F4/80^+^ MPs; (**d**) Western blot of Nfix expression in WT MPs treated with DMSO (Dimethyl sulfoxide) or Y27632 for 16 h and quantification. Vinculin was used to normalize; (**e**) WT MPs were treated with DMSO or Y27632 for 16 h and were immunolabeled for pro-inflammatory markers (iNOS and TNFα) and anti-inflammatory markers (TGFβ and CD163). The number of positive cells is expressed as percentage out of total cells; (**f**) LysM^CRE^:Nfix^fl/fl^ MPs were treated with DMSO or Y27632 for 16 h and were immunolabeled for pro-inflammatory markers (iNOS and TNFα) and anti-inflammatory markers (TGFβ and CD163). The number of positive cells is expressed as percentage out of total cells. * *p* < 0.05, *** *p* < 0.001. Results are means ± SEM of at least three independent experiments.

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
