# Peer review of "The Transcription Factor Nfix Requires RhoA-ROCK1 Dependent Phagocytosis to Mediate Macrophage Skewing during Skeletal Muscle Regeneration"

_cells, 2020, doi:10.3390/cells9030708_

Round 1

Reviewer 1 Report

The paper presented by Saclier and colleagues aims to demonstrate that RoA-ROCK1 dependent phagocytosis is required to mediate Nfix action on macrophages during muscle regeneration, and in particular to induce their switch from pro- to anti-inflammatory phenotype. 

The paper is carefully written, and easy to follow. However, there are some issues.

1) Have you tried to check the expression of IL4 and IL6 during muscle regeneration in both mouse models?

2) LysM drives the expression of Cre not only in MPs, but also in polymorphonuclear cells, such as neutrophils. Neutrophils are known to be recruited at injury site very rapidly, even before macrophages. Nfix is expressed also in neutrophils, so is it possible that they play a role in the delayed regeneration process, observed in the LysMCre:Nfixfl/fl mice?

3) It is stated that two-tailed unpaired Student's t-Test was used. It does not seem that this kind of statistic test is appropriate for all the experiments presented. In Figure 1a, for example, a one-way ANOVA would be more appropriate, while a two-way ANOVA should be used to analyze data in Fig. 1b. The same is true for Fig 2c, or Fig 4. 

Author Response

We would like to thank the Editor and Reviewers for careful reading of this manuscript and for the positive and constructive suggestions, which help to improve the quality of this work.

Listed below are our point-to-point replies to the 1st Reviewer’s comments. Modifications in the manuscript appear in blue.

Review 1

The paper presented by Saclier and colleagues aims to demonstrate that RoA-ROCK1 dependent phagocytosis is required to mediate Nfix action on macrophages during muscle regeneration, and in particular to induce their switch from pro- to anti-inflammatory phenotype. 

The paper is carefully written, and easy to follow. However, there are some issues.

We thank the Reviewer for her/his interest in our work

1) Have you tried to check the expression of IL4 and IL6 during muscle regeneration in both mouse models?

We thank the Reviewer 1 for raising this point. We did not test these two cytokines.  We started our study by looking at the phenotype of macrophages lacking Nfix in vitro by immunofluorescence (3 markers for pro-inflammatory phenotype and 4 for anti-inflammatory phenotype, Figure 3c), demonstrating a defect of anti-inflammatory phenotype acquisition in macrophages lacking Nfix and in which nor IL4 neither IL6 were used. After muscle injury, we observed in vivo that the skewing does not occur properly. (Figure 3a and b). We therefore decided to mainly focus our analysis on the Ly6C+ to Ly6C- switch and on how this might impinge myogenic features upon acute injury, rather than the expression of pro- and anti-inflammatory markers. The rationale for that is because a study demonstrated that simply identification of defined markers is not useful to discriminate macrophage activation status (Mounier et al, 2013). Surely, future analysis might reveal a persistence of IL6 expression and decrease/delayed of IL4 expression could occur in the damaged muscle of LysMCre:Nfixfl/fl mice compared to the Nfixfl/fl mice.

2) LysM drives the expression of Cre not only in MPs, but also in polymorphonuclear cells, such as neutrophils. Neutrophils are known to be recruited at injury site very rapidly, even before macrophages. Nfix is expressed also in neutrophils, so is it possible that they play a role in the delayed regeneration process, observed in the LysMCre:Nfixfl/fl mice?

We thank the Reviewer 1 for this question. To our knowledge, the expression of Nfix in neutrophils has never been published. The reason why we did not look at this population is the following. Neutrophils rapidly infiltrate cardiotoxin damaged muscle and reach a pic between 1-2 days after damage and then are no more present anymore in injured muscle at 4 days (Tidball and Villalta, 2010; Mounier et al, 2013; Hardy et al, 2016). In the study published in Cell Metabolism, Mounier et al observed that in the total of CD45+ cells present in the muscle 2 days after cardiotoxin injection, more than 2/3 are macrophages and that at 4 days after injection almost all macrophages present within the muscle are anti-inflammatory (Mounier et al, 2013). In our study, we do not observed a morphological difference between the Nfixfl/fl and the LysMCre:Nfixfl/fl mice at 2 days upon cardiotoxin injection but at 4 days (Figure 2a and b). The in vivo experiments also showed that both phenotype and function of anti-inflammatory macrophages, but not pro-inflammatory macrophages, on myogenic cells are compromised in absence of Nfix (Figure 3c, Figure 4a and b). We are therefore convinced that the delayed muscle regeneration observed in LysMCre:Nfixfl/fl mice is mainly caused by lack of Nfix in macrophages.

3) It is stated that two-tailed unpaired Student's t-Test was used. It does not seem that this kind of statistic test is appropriate for all the experiments presented. In Figure 1a, for example, a one-way ANOVA would be more appropriate, while a two-way ANOVA should be used to analyze data in Fig. 1b. The same is true for Fig 2c, or Fig 4. 

We thank the Reviewer 1 for this remark. We now used statistical test as suggested by the Reviewer 1. For the Figure 1a, using a one-way ANOVA we showed the same results (p<0,001). For Figure 1b, we also obtained same results after using ANOVA two-way, but we observed a mistake in the legend that now we corrected. For Figure 2c at D14, we indeed obtained a p<0,001 for myofibers having area between 0-500μm2, so we changed both figure and figure legend. On the contrary, we did not observed any difference at D28 for CSA >5000μm2, here again we change the figure 2c. For Figure 4, we obtained the same results. We change the 2.11 Statistical Analysis in Materials and Methods according to the new statistical tests realized.

Reviewer 2 Report

In this study the authors have investigated the role of the transcription factor Nfix during skeletal muscle regeneration. They reported that Nfix is the link between phagocytosis and the phenotypical switch of macrophages, a necessary step for the resolution of the inflammation and skeletal muscle regeneration.

This paper is very interesting and the experiments support the conclusions. However, I have some criticisms.

Criticisms:

1- Line 319-320, the authors showed that conditioned medium increases the fusion index of myogenic progenitor cells. They found a fusion index of 30-50% (figure 4) but I am very surprised by the low number of cells and myotubes (figure S4). Why are there so few cells?  

2-For the quantification of the CSA of myofibers, how was the analysis done? All fibers have been counted? Or just some fields? This is important because in the tibialis the fibers type is heterogeneous?

3-In the discussion (line 422), the authors stated that the absence of Nfix leads to muscle regenerative defect. Is it a defect or just a delay in the regeneration process? The regeneration process might be incomplete rather that deficient in these mice. What happen 2 months after cardiotoxin injection in Nfix deficient mice?

4-line 225: Nfix is mainly express by inflammatory MPs. “Mainly” is exaggerated, “more” will be fairer.   

Author Response

We would like to thank the Editor and Reviewers for careful reading of this manuscript and for the positive and constructive suggestions, which help to improve the quality of this work.

Listed below are our point-to-point replies to the 1st Reviewer’s comments. Modifications in the manuscript appear in blue.

Review 2

In this study the authors have investigated the role of the transcription factor Nfix during skeletal muscle regeneration. They reported that Nfix is the link between phagocytosis and the phenotypical switch of macrophages, a necessary step for the resolution of the inflammation and skeletal muscle regeneration.

This paper is very interesting and the experiments support the conclusions. However, I have some criticisms.

We thank the Reviewer for her/his interest in our study. 

Criticisms

1- Line 319-320, the authors showed that conditioned medium increases the fusion index of myogenic progenitor cells. They found a fusion index of 30-50% (figure 4) but I am very surprised by the low number of cells and myotubes (figure S4). Why are there so few cells?  

We thank the Reviewer for the remarks. We modified the Figure S4 and replaced images of mpcs and Nfixfl/fl M2c by more representative pictures.

2-For the quantification of the CSA of myofibers, how was the analysis done? All fibers have been counted? Or just some fields? This is important because in the tibialis the fibers type is heterogeneous?

For the quantification of CSA, we analyzed those damaged TA which presented at least 75% of injured muscle. We then randomly took at least 8 pictures in different fields and we always analyzed at least 500 myofibers.  

3-In the discussion (line 422), the authors stated that the absence of Nfix leads to muscle regenerative defect. Is it a defect or just a delay in the regeneration process? The regeneration process might be incomplete rather that deficient in these mice. What happen 2 months after cardiotoxin injection in Nfix deficient mice?

We thank the Reviewer 2 for raising this point that, actually, needed a clarification. In mice lacking Nfix in macrophages, we analyzed the phenotype of damaged muscles until 1 month upon injury. In these mice, we observed a reformation of myofibers but the general regenerative process is delayed. As suggested by the Reviewer 2, we changed defect by delayed in the discussion. Since we observed a defect of phenotypical skewing between D2 and D4 after cardiotoxin injury, we decided to focus on that phenomena and looked at that defined temporal window.

As at our knowledge, few studies on genetically modified macrophages in models of muscle regeneration looked up to 2 months upon cardiotoxin injection.

4-line 225: Nfix is mainly express by inflammatory MPs. “Mainly” is exaggerated, “more” will be fairer.   

We thank the Reviewer 2 and modify the text as suggested.
